# A Simple Method to Identify the Dominant Fouling Mechanisms during Membrane Filtration Based on Piecewise Multiple Linear Regression

**DOI:** 10.3390/membranes10080171

**Published:** 2020-07-29

**Authors:** Hao Xu, Kang Xiao, Jinlan Yu, Bin Huang, Xiaomao Wang, Shuai Liang, Chunhai Wei, Xianghua Wen, Xia Huang

**Affiliations:** 1College of Resources and Environment, University of Chinese Academy of Sciences, Beijing 100049, China; xuhao9510@163.com (H.X.); yujinlan18@mails.ucas.ac.cn (J.Y.); 2School of Civil Engineering, Guangzhou University, Guangzhou 510006, China; huangbinstudy@163.com; 3Center for Ocean Mega-Science, Chinese Academy of Sciences, Qingdao 266071, China; 4State Key Joint Laboratory of Environment Simulation and Pollution Control, School of Environment, Tsinghua University, Beijing 100084, China; wangxiaomao@tsinghua.edu.cn (X.W.); xhwen@tsinghua.edu.cn (X.W.); xhuang@tsinghua.edu.cn (X.H.); 5College of Environmental Science and Engineering, Beijing Forestry University, Beijing 100083, China; shuai_liang@bjfu.edu.cn; 6Research and Application Center for Membrane Technology, School of Environment, Tsinghua University, Beijing 100084, China

**Keywords:** fouling development model, filtration law, pore blocking, multiple linear regression, statistical test

## Abstract

Membrane fouling is a complicated issue in microfiltration and ultrafiltration. Clearly identifying the dominant fouling mechanisms during the filtration process is of great significance for the phased and targeted control of fouling. To this end, we propose a semi-empirical multiple linear regression model to describe flux decline, incorporating the five fouling mechanisms (the first and second kinds of standard blocking, complete blocking, intermediate blocking, and cake filtration) based on the additivity of the permeate volume contributed by different coexisting mechanisms. A piecewise fitting protocol was established to distinguish the fouling stages and find the significant mechanisms in each stage. This approach was applied to a case study of a microfiltration membrane filtering a model foulant solution composed of polysaccharide, protein, and humic substances, and the model fitting unequivocally revealed that the dominant fouling mechanism evolved in the sequence of initial adaptation, fast adsorption followed by slow adsorption inside the membrane pores, and the gradual growth of a cake/gel layer on the membrane surface. The results were in good agreement with the permeate properties (total organic carbon, ultraviolet absorbance, and fluorescence) during the filtration process. This modeling approach proves to be simple and reliable for identifying the main fouling mechanisms during membrane filtration with statistical confidence.

## 1. Introduction

Membrane fouling is an obstinate problem in microfiltration (MF, see Appendix C for all the Abbreviations) and ultrafiltration (UF) for water and wastewater treatment [1,2,3,4,5]. Membrane fouling refers to the process of flux decline (or increase in hydraulic resistance) during filtration due to the deposition of suspended or soluble substances on the membrane surface, at the pore openings, or inside the pores. As a result, membrane fouling reduces the efficiency of filtration. Membrane fouling includes organic fouling, inorganic fouling, and biofouling. Organic matter does not only cause organic fouling which is prevalent in MF and UF systems; it is also seriously involved in inorganic–organic combined fouling (such as impervious gel layers due to metal–organic complexation) and bio-organic fouling (such as biofilms stubborn against cleaning). Among organic foulants, polysaccharides, proteins, and humic acids are the most reported [6,7]. Upon removing the foulants from the membrane, frequent physical or chemical cleaning will increase energy consumption, shorten membrane life and increase operating costs. A realistic goal of fouling control is to preventively delay and mitigate fouling or to more efficiently remove the deposited foulants in a well phased and targeted manner [8,9,10]. To this end, it is of great significance to clearly identify the key mechanisms for the dynamic development of membrane fouling, from internal pore blockage to external cake build-up, using appropriate analytical or modeling approaches.

Classical filtration laws have been widely used to describe the fouling process. The classical filtration laws include four basic types of fouling, which can be expressed by a collective expression:(1)$$\frac{d^{2}t}{dV^{2}}=k{\left(\frac{dt}{dV}\right)}^{N}\mathrm{or}\;\mathrm{equivalently},\;\frac{dR}{dV}=kR^{N}$$
where *t* is filtration time, *V* is specific permeate volume, *R* is filtration resistance, *k* is a model coefficient, and the characteristic exponent *N* indicates fouling modes: *N* = 2, 1.5, 1, and 0 represent complete blocking (clogging at the pore opening), standard blocking (the first kind for instantaneous adsorption of foulant on the pore wall), intermediate blocking (random deposition on the membrane surface) and cake filtration (uniform growth of a cake or gel layer on the surface), respectively. Recently, the standard blocking law has been extended to include the second kind (*N* = 2.5) for the case of slow adsorption (e.g., hydrophilic foulant on hydrophilic pore walls) compared to permeate advection through the pores [11]. Mathematical modeling plays an essential role in interpreting various fouling mechanisms and processes. Through model fitting using Equation (1), it is plausible that the characteristic exponent N, and, hence, the type of fouling mechanism can be determined.

However, the actual situation is often more complicated. First, a variety of mechanisms may coexist. Regarding this, researchers have developed combined models incorporating different fouling types. Many researchers have derived dually combined fouling models such as the cake-complete blocking, cake-intermediate blocking, cake-standard blocking, complete-standard blocking, and intermediate-standard blocking models [12,13,14,15,16,17,18,19]. Some have combined three or even four mechanisms to interpret fouling behavior. Duclos-Orsello et al. developed a flux decline model that combined three fouling mechanisms (standard blocking, complete blocking, and cake filtration) to describe the MF process of a bovine serum albumin solution [20]. Kim et al. combined all the four classical fouling modes into one mechanistic model to explain the effect of coexisting fouling mechanisms [21]. However, the forms of the combined models are often complex (the more combinations, the more complex), thus requiring nonlinear regression [22] and even genetic algorithms for the model fitting [21]. Second, the main mechanism may vary largely during the filtration process. However, a large number of researchers have used a fixed model to fit the whole process [23,24], and only a few have conducted piecewise fitting using multiple models [12,25,26]. The unreasonable setting of the time range or the number of time segments in the model fitting inevitably lead to deviations, such as the resultant autocorrelation of time series that has seldom been mentioned in previous fouling studies. 

In the present study, we propose a simple and straightforward multiple linear regression model to describe the piecewise succession of five pore-blocking/cake-filtration fouling mechanisms during the full filtration process, based on the idea of the linear additivity of filtration volumes under different fouling mechanisms. The filtration curve (flux vs. volume) was fitted for each time segment, and the stepwise regression technique was used to screen out insignificant mechanisms and to identify the dominant mechanism in each time segment. The Durbin–Watson (DW) autocorrelation test was employed to justify the most reliable setting of the number and length of the time segments [27]. By using the proposed method, one can easily access the profile of fouling mechanisms during the filtration process with statistical confidence. In comparison with the previous models, this method has the merits of the more complete inclusion of possible mechanisms, the more complete coverage of fouling stages, the more realistic handling of complex fouling caused by multiple foulants, and being statistically more rigorous. This method will help evaluate membrane potential, characterize foulant properties, and understand membrane–foulant interactions. MATLAB code is provided to support convenient usage of this method.

## 2. Model 

The expression of the classical filtration laws, as shown in Equation (1), can be transformed to yield the flux vs. filtration volume relationship:(2)$$\frac{dJ}{dV}=-kJ^{2-N}$$
where *J* is the filtration flux (*J* = d*V*/d*t*). The filtration curve can be described by five filtration models with *N* = 2.5, 2, 1.5, 1, and 0, as shown schematically in Figure 1. By integrating Equation (2) on both sides, the *J*–*V* relationships for these models are obtained as follows: (3)$$V-V_{0}=\{\begin{array}{cc} \begin{array}{l} k_{1}(J_{0}{}^{0.5}-J^{0.5})\\ k_{2}(J_{0}{}^{1.5}-J^{1.5})\\ k_{3}(J_{0}-J)\\ k_{4}(\ln J_{0}-\ln J)\\ k_{5}(J^{-1}-J_{0}{}^{-1}) \end{array} & \begin{array}{l} (1\mathrm{st}-\mathrm{kind}\;\mathrm{standard}\;\mathrm{blocking},\;N=1.5)\\ (2\mathrm{nd}-\mathrm{kind}\;\mathrm{standard}\;\mathrm{blocking},\;N=2.5)\\ (\mathrm{complete}\;\mathrm{blocking},\;N=2)\\ (\mathrm{intermediate}\;\mathrm{blocking},\;N=1)\\ (\mathrm{cake}\;\mathrm{filtration},\;N=0) \end{array} \end{array}$$
where the subscript “0” refers to the initial state.

It is possible that different fouling mechanisms can simultaneously occur at different positions of the membrane. Assuming that *f_i_* is the proportion of membrane area occupied by the *i*th mechanism and *V_i_* is the resultant specific permeate volume, the contribution of various possible mechanisms to the total *V* can be expressed by *V* = ∑*f_i_V_i_* [28], as also illustrated in Figure 1. By substituting *V_i_* using Equation (3), a comprehensive model of membrane fouling can thus be established in the form of:(4)$$V=\sum f_{i}V_{i}=k_{0}+k_{1}(-J^{0.5})+k_{2}(-J^{1.5})+k_{3}(-J)+k_{4}(-\ln J)+k_{5}(J^{-1})+\varepsilon$$
where *V* is the specific permeate volume of the whole membrane, *J* is the membrane flux of the whole membrane, *k*_0_ is a constant term, and *k*_1_, *k*_2_, *k*_3_, *k*_4_, and *k*_5_ are the coefficients of 1st kind of standard blocking (fast adsorption), the 2nd kind of standard blocking (slow adsorption), complete blocking, intermediate blocking, and cake filtration, respectively. Note that the original *k_i_*’s in Equation (3) and the *f_i_*’s were incorporated into the *k_i_*’s in Equation (4). The error term, *ε*, encompasses errors due to the random error of *V* and *J*, the deviation of *N* (as a result of, e.g., non-uniformity of the pore structure [11,28,29] or polydispersity of the foulant particles [30]), the neglect of concentration polarization (a feedback effect on foulant mass transfer), and other fouling mechanisms [7,28,31], as well as possible interactive influences between different mechanisms (e.g., at adjacent areas of the membrane) that cause the nonlinearity of Equation (4).

Taking *V* as the dependent variable and the five *J*-derived terms as independent variables, Equation (4) can be used to fit the experimental *J*–*V* data via least-squares regression. However, even though the multiple linear regression model takes account of all the five mechanisms, the fitting may still fail because the proportions (*f_i_* or *k_i_*) of the mechanisms may change and the dominant mechanism(s) may shift during the filtration process. Therefore, a single regression model only holds in a time segment during which the mechanisms do not change significantly. In this context, it is more realistic to perform piecewise fitting into each time segment respectively, rather than fit the same model across the whole period. In each segment, the fitted result is validated by two criteria: (a) All significant coefficients (any of *k*_1_–*k*_5_) should be positive, and (b) the autocorrelation of time series residuals should not be significant.

The significance of the coefficients is judged by the *t*-test (by the convention that *p* < 0.05 means significant), and a significantly positive *k_i_* means that the *i*th mechanism is significant in this segment. The Durbin–Watson (DW) test is employed to judge the autocorrelation of time series residuals, i.e., the correlation between the regression residuals of two adjacent time points. In linear regression, it is always assumed that the residuals are independent (uncorrelated). If the assumption of independence is violated, the fitting results become problematic. For example, the positive correlation between error terms tends to amplify the value of coefficient, making the predicted variables appear important, even though they may, in fact, not be important. When fitting experimental data using Equation (4), autocorrelation arises from the unreasonable division of the time span when (a) the time segment is too long, which sacrifices the local accuracy of model fitting, or (b) the time segment is shorter than the segment used for flux calculation (*J* = ∆*V*/∆*t*), which causes periodic oscillation of the residuals. The autocorrelation is deemed to be significant when the *p* value of the DW test (*p*_DW_) is smaller than 0.05.

The protocol for optimizing the piecewise fitting is as follows: (a) evenly divide the filtration process *V* into *n* parts (*n* starts from 1), (b) conduct least-squares regression in the “stepwise” mode in each segment to automatically screen out insignificant *k_i_*’s, and check the DW test results for the remained significant *k_i_*’s, and (c) increase *n* and repeat the regression until the remained *k_i_*’s are all significantly positive and the DW test is not significant for all of the segments. 

The MATLAB codes for raw data processing (to calculate *J*, *R*, and d*R*/d*V* automatically from the original *t* and *V* data) and for the piecewise fitting are provided in Appendix A and Appendix B, respectively.

## 3. Experimental

### 3.1. Membrane and Model Foulant Solution

Polyvinylidene fluoride (PVDF) flat-sheet membranes with a nominal pore size of 0.1 μm (VVLP, Millipore, MA, USA) were employed in the fouling experiments. Prior to use, the membranes were rinsed and immersed in Milli-Q water for 24 h, and then they were kept in a salt background solution for 24 h to eliminate soluble impurities. The salt background solution consisted of 2 mM CaCl_2_, 1 mM MgCl_2_, 2 mM NaHCO_3_, 12 mM NaCl, and 0.2 mM Na_2_SiO_3_ in accordance with the salt background solution of the model foulant solution used in the fouling experiments. 

The model foulant solution was comprised of 16 mg/L sodium alginate (SA) (MACKLIN, Shanghai, China), 8 mg/L humic acid (HA) (Aladdin, Shanghai, China), 4 mg/L bovine serum albumin (BSA) (Sigma-Aldrich, Saint Louis, MO, USA), and the aforementioned salt background. The polysaccharides/humics/proteins proportion was similar to that of membrane bioreactor supernatants in municipal wastewater treatment [32]. The solution was prepared by: (a) dissolving the organic foulant components in a 75% volume of Milli-Q water followed by 12 h of stirring, (b) adding the salts with the other 25% volume of Milli-Q water followed by another 12 h of stirring, and (c) filtering through a glass-fiber membrane (0.7 µm, GF/F, Whatman, Maidstone, UK) to remove undissolved coarse particles from the liquid. 

### 3.2. Filtration Test

A dead-end filtration system was employed for the fouling development test at room temperature and constant pressure. The system consisted of a nitrogen cylinder, a pressure regulating value, a liquid storage tank, a filtration cell (Amicon 8400, Millipore, MA, USA) with an effective filtration area of 41.8 cm^2^ and a volume of 400 mL, an electronic balance (PL2002, Mettler Toledo, Zurich, Switzerland), and a computer, as shown in Figure 2. The fouling test was carried out in the procedure of: (a) pre-pressing the membrane under 5 kPa for 1 h to stabilize the deformation caused by pressure, (b) pre-filtering 200 mL of the salt background solution to measure the initial filtration resistance of the membrane, (c) pre-contacting the membrane with the model foulant solution at 5 kPa for 1 h without filtration to adapt the physicochemical state of the membrane to the solution, and (d) filtering the model foulant solution at 5 kPa with the water level in the filtration cell maintained at 200 mL (the water level was maintained via pressure balance and the net volume filtered was supplemented by the storage tank). The real-time filtration resistance *R* (m^−1^) was calculated according to Darcy’s law:(5)$$R=\frac{P}{\mu J}=\frac{P}{\mu }{\left(\frac{\Delta V}{\Delta t}\right)}^{-1}$$
where *P* is the trans-membrane pressure (Pa), *μ* is the dynamic viscosity of the permeate (Pa s, approximately that of water) with the effect of temperature corrected, and *J* is the real-time filtration flux (m/s) calculated as ∆*V*/∆*t*, where *V* is the specific permeate volume (m^3^/m^2^). For the calculation of *J*, the time increment ∆*t* was found given a fixed ∆*V* of 0.002 m^3^/m^2^ (i.e., 0.2 cm^3^/cm^2^); see Appendix A for details.

### 3.3. Analytical Items

The total organic carbon (TOC) concentrations in the feed and permeate were determined using a TOC analyzer (Multi N/C 3100, Jena, Germany). The ultraviolet–visible (UV–Vis) absorbance of the foulants was measured using a spectrophotometer (T3200, YOKE, Shanghai, China) with a scanning range of 200–500 nm. The fluorescence signals of the permeate were scanned using a fluorescence spectrophotometer (Cary Eclipse, Agilent, Palo Alto, CA, USA) in the 3D mode over the excitation wavelength range of Ex = 200–400 nm and the emission wavelength range of Em = 250–500 nm. The obtained excitation–emission matrix (EEM) spectra were treated according to the following steps [33]: (a) subtracting pure water background, (b) removing Rayleigh and Raman scatterings, (c) correcting the inner-filter effect using UV–Vis absorbance, and (d) standardizing the fluorescence intensity (FI) into Raman units (R.U.) using the Raman signal of pure water as a reference. 

## 4. Results and Discussion

### 4.1. Overview of Fouling Evolution

The development of membrane fouling can be reflected by the change of membrane flux (*J*) and filtration resistance (*R*) with specific permeate volume (*V*). *J* and *R* were automatically calculated from the raw *t* and *V* data at an interval of ∆*V* = 0.2 cm^3^/cm^2^, using the homemade MATLAB function “VJR,” as given in Appendix A. As can be seen from Figure 3, *R* increased slowly when *V* was smaller than 7 cm^3^/cm^2^, and it increased rapidly thereafter. There are several turning points in the *J*–*V* curve (such as those roughly around *V* = 2, 7 and 9 cm^3^/cm^2^), implying that there must have been more than one fouling mechanism over the whole process and the temporarily dominant mechanism was likely to change with *V*. However, it is difficult to accurately distinguish different fouling stages of the whole filtration process by only referring to the *R*–*V* or *J*–*V* curves. The further analysis of the *R*–*V* or *J*–*V* relationship is required. Note that the seven stages marked in Figure 3, as well as the fitted curve, were obtained by piecewise regression, as is described later in detail.

A classical method for examining a fouling mechanism (e.g., blocking mode) is to plot the ln(d*R*/d*V*)~ln*R* data and determine the slope of the curve as the characteristic *N* value (Equation (1)). In Figure 4, ln(d*R*/d*V*) is shown to have decreased slightly at the very beginning of the filtration (stage 1), increased almost linearly with ln*R* in the middle stages of 2–5, and gradually reached a plateau after prolonged filtration (stages 6–7). Note that the so-termed seven stages were distinguished according to piecewise regression, as is introduced later. The plateau (i.e., *N* = 0) indicates that a cake or gel layer was eventually formed on the membrane surface. However, the large fluctuation of the data points rendered a large uncertainty of slope calculation. The fluctuation was very likely due to that the original *V*~*t* data that had been derived for many times to obtain d*R*/d*V*, and this concern has also been seriously raised by a number of researchers [12,31]. Considering that there were five possible mechanisms, we tentatively fitted the ln(d*R*/d*V*)~ln*R* data using a fifth-order polynomial trend line. Monte Carlo simulation based on the distribution of fitting residuals showed that the 95% confidence interval of the trend line was rather wide (Figure 4). Therefore, there was great uncertainty in judging the mechanisms with *N* value from the slope of the trend line, especially in the rising section. Alternatively, a better choice may have been fitting the *J*–*V* (or *R*–*V*) rather than d*R*/d*V*–*R* data to reduce the uncertainty by using a piecewise fitting approach to deal with the changing mechanisms over the stages. 

### 4.2. Piecewise Fitting Results

The whole filtration process (*V* ≈ 12 cm^3^/cm^2^) was evenly divided into *n* segments (*n* = 1, 2, …), and the segmented *J*–*V* data were subjected to piecewise fitting of the multiple linear model (Equation (4)) using the homemade MATLAB function “StepwiseModel,” as given in Appendix B. The appropriateness of the division was judged by a *t*-test for the significance of positive *k_i_* coefficients and the DW test for the independence of the time series residuals. These tests were not successful until the segment number *n* increased to 7. The seven stages are marked in Figure 3, where it can be seen that the piecewise fitted *J*–*V* curve matched closely to the experimental data. As a consequence of the piecewise fitting, the significant *k_i_* coefficients, as well as the DW test results along the seven stages, are shown in Figure 5. The *p*_DW_ values were no smaller than 0.05, suggesting that the autocorrelations of residuals were tolerable. From the *k_i_* coefficients, it is evident that the dominant fouling mechanism evolved in a sequence of: cake filtration (difficult to explain, possibly due to sudden change of hydraulic state at the beginning of filtration) (stage 1) → fast adsorption inside the membrane pores (stages 2–3) → slow adsorption inside the membrane pores (stages 4–5) → gradual growth of a cake layer on the membrane surface (stages 6–7).

In comparison with the piecewise multiple linear regression results, the apparent *N* values were calculated using the conventional approach, i.e., from the slope of the ln(d*R*/d*V*)–ln*R* curve in Figure 4, with their standard deviations estimated via Monte Carlo simulation. The results are shown in Figure 5. It is obvious that the apparent *N* values exhibited great uncertainty due to the large fluctuation of the ln(d*R*/d*V*)–ln*R* data. The uncertainty made some of the *N* values deviate widely from the theoretical values. The deviation might have also been due to coexistence of multiple fouling mechanisms. For example, at the first kind of standard blocking-dominated stage 3, the apparent *N* of around 2 might have partly been due to coexistence of the emerging second kind of standard blocking; at the second kind of standard blocking-dominated stage 4, the apparent *N* of larger than 2.5 might have partly been due to the noncircularity of the actual membrane pores [11] or the acceleration of pore blocking efficiency when the adsorbed foulant concentration was approaching the gel point to form microgels in the pore channels. These uncertainties rendered the apparent *N* unreliable for identifying the main mechanism(s) during the filtration process. The apparent *N* value was sometimes even misleading; for instance, a value of 2 seemingly pointed to a single mechanism of complete blocking, whereas in fact it might have been a weighted average of two other mechanisms (e.g., first and second kinds of standard blocking with the true *N* values of 1.5 and 2.5, respectively).

In contrast to the apparent *N*, the piecewise multiple linear regression model could unequivocally identify the major mechanism(s) along the fouling stages with statistical confidence. The model was conservative in terms of the five known mechanisms, as it incorporated any uncertain factors (such as the aforementioned nonideality of membrane pores and fouling process) collectively into the error term (Equation (4)). The relative importance (signal-to-noise ratio) of the main mechanism(s) compared to the error term was finally judged by statistical tests (*F*-test of the model or *t*-test of the coefficients). Therefore, this approach is considered to be more robust than the conventional apparent *N* approach.

### 4.3. Interpreting the Transition of Major Mechanisms According to Foulant Composition

The modeling results were further evidenced by the consecutive measurements of the permeate properties during the filtration process. The TOC measured the total dissolved organic matter (DOM), UV absorbance at 254 nm (UV_254_) indicated the chromophoric portion of the DOM such as proteins and other unsaturated components [34,35], and the averaged fluorescence intensity at the emission wavelength range of 380–500 nm (FI_380–500_) mainly reflected the humic components of the DOM [33,36]. In Figure 6, the permeate TOC and UV_254_ both show a “Z”-shaped decreasing trend with the increase of *V*. At stages 5–6, the TOC dropped sharply, indicating that the overall rejection rate of organic foulants rose sharply due to the formation of a cake/gel layer on the membrane surface. At stage 7, the permeate TOC became stable at a low level of 2–3 mg/L, indicating full coverage and the steady growth of the cake/gel layer on the membrane surface. Correspondingly, the TOC rejection rate was below 10% at the pre-cake stages (stages 1–4), but it grew to over 60% at the cake/gel layer stage (stage 7). The cake/gel layer was mainly constructed of alginate since alginate was the major TOC contributor in the model foulant solution. The same trend of UV_254_ suggested that the chromophoric portion of the DOM (such as protein) also participated in the formation of the cake or gel layer. In contrast to the TOC and UV_254_, the FI_380–500_ exhibited an “S”-shape increase during the filtration process, reflecting the gradual breakthrough of some small fluorescent molecules (such as some humic species) through the membrane and the cake/gel layer due to the equilibration of the dynamic adsorption [37,38]. The above showed that polysaccharide (alginate) and protein fouled the membrane mainly through forming a surface cake/gel layer (most likely polysaccharide formed the gel network and protein was trapped in it [7]), while small molecular humic substances successively underwent fast and slow adsorption, and they finally penetrated through the membrane and cake/gel layer. This is basically consistent with the understanding in the literature [39,40]. All the effluent properties were in good agreement with the model fitting results, which proves the rationality of the method proposed in this paper in identifying the main fouling mechanism during the membrane filtration process.

This comprehensive model, encompassing five mechanisms for pore blockage and cake filtration, should be widely applicable to various situations of dead-end filtration. For example, when the particle size is much smaller than the pore size, standard blocking may be more dominant in the model. In the case of small particle size and weak hydrophobicity, the blockage may fall into the regime of the second kind of standard blocking, which is slow but lasts long. If the particle size is comparable to the pore size, complete blocking may rapidly occur. Moreover, the evolution to gel layer stage may be postponed or advanced given different hardness ion concentrations for metal–organic complexing gel layer formation. These situations are all within the scope of the comprehensive model.

## 5. Conclusions

In this paper, we proposed a semi-empirical multiple linear regression model that incorporates the five fouling mechanisms (first and second kinds of standard blocking, complete blocking, intermediate blocking, and cake filtration). MATLAB codes are provided for data processing and model fitting (“VJR” and “StepwiseModel” in the Appendices), which enable the optimal segmentation of the filtration process for the piecewise regression and refining of significant parameters with statistical tests. This provides a simple and rigorous method for identifying the main fouling mechanisms during the filtration process. In the case study of a Millipore MF membrane fouled by a model solution with polysaccharide, protein, and humic components, the model fitting results showed that the dominant fouling mechanism evolved in the order of: initial adaptation (stage1) → fast adsorption inside the membrane pores (stages 2–3) → slow adsorption inside the membrane pores (stages 4–5) → gradual growth of a cake/gel layer on the membrane surface (stages 6–7). The model fitting results were in good agreement with the permeate properties during the filtration process, which proves the rationality and effectiveness of this method in fouling mechanism study. It also provides a tool to assess membrane fouling potential, characterize foulant properties, and understand membrane–foulant interactions, all of which will profoundly support optimal selection of membrane and targeted pretreatment of foulant solution for efficient fouling control in industrial applications.

## Figures and Tables

**Figure 1 membranes-10-00171-f001:**
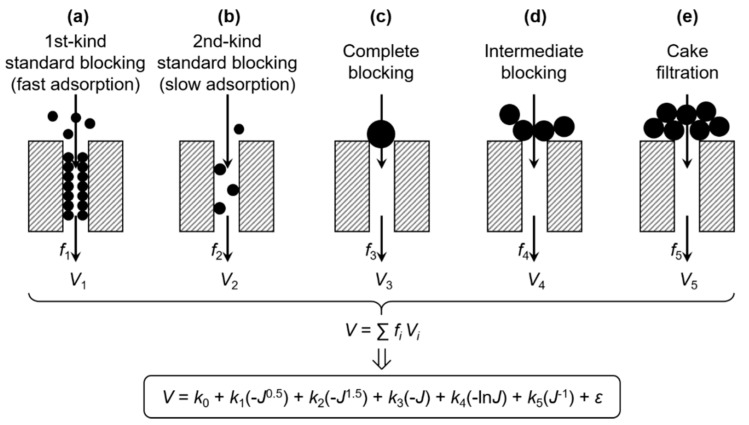
Scheme of the multiple linear regression model for membrane fouling.

**Figure 2 membranes-10-00171-f002:**
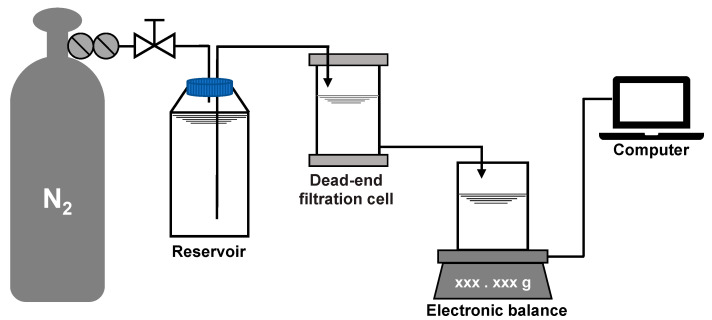
Layout of the dead-end filtration system.

**Figure 3 membranes-10-00171-f003:**
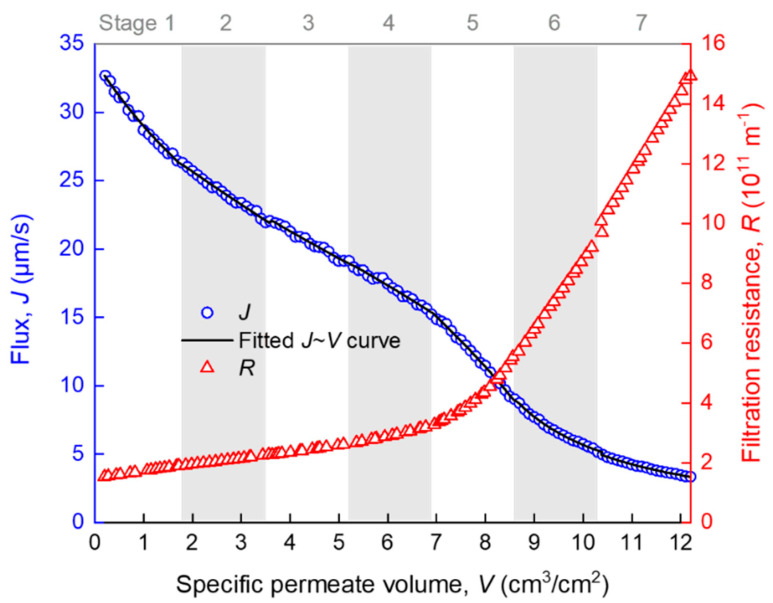
Variation of membrane flux and filtration resistance during the filtration process. Data are given as average from two repeated experiments with a relative error smaller than 4.4% (the error bars are thus omitted for simplicity of presentation). The R-squared values for the fitting in stages 1–7 were 0.983, 0.990, 0.985, 0.983, 0.996, 0.980, and 0.987, respectively.

**Figure 4 membranes-10-00171-f004:**
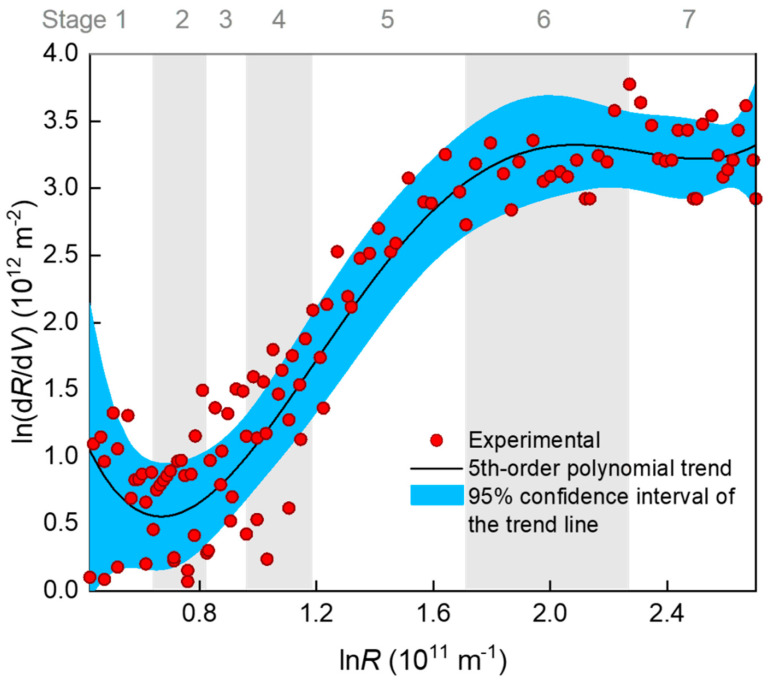
Relationship between d*R*/d*V* and *R* (filtration resistance) during the filtration process. The R-squared value of the 5th order polynomial fitting was 0.907. The 95% confidence interval of the 5th order polynomial trend line was produced using a 999-times Monte Carlo simulation based on the distribution of residuals.

**Figure 5 membranes-10-00171-f005:**
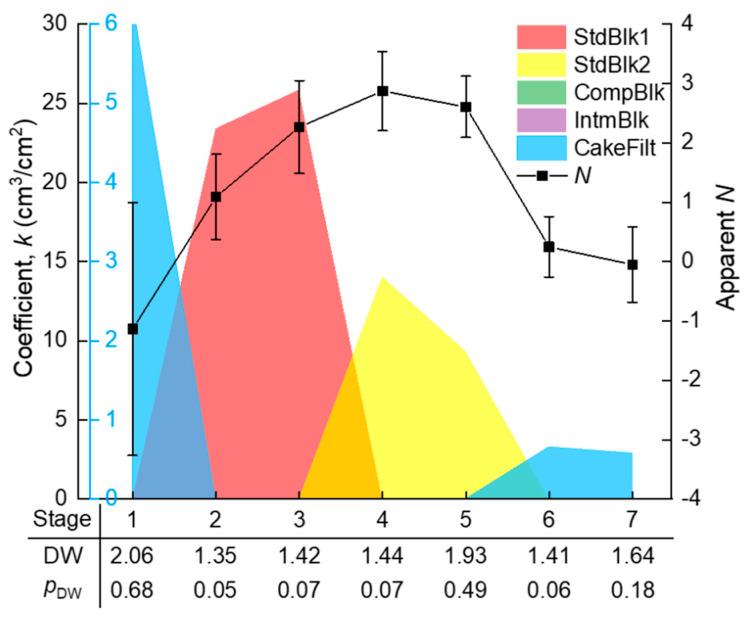
The significant coefficients and the Durbin–Watson (DW) test results given by piecewise fitting of the filtration flux~specific permeate volume (*J*–*V)* data in Figure 3, and the apparent *N* (fouling modes) values (± standard deviation) calculated from the ln(d*R*/d*V*)–ln*R* curve in Figure 4.

**Figure 6 membranes-10-00171-f006:**
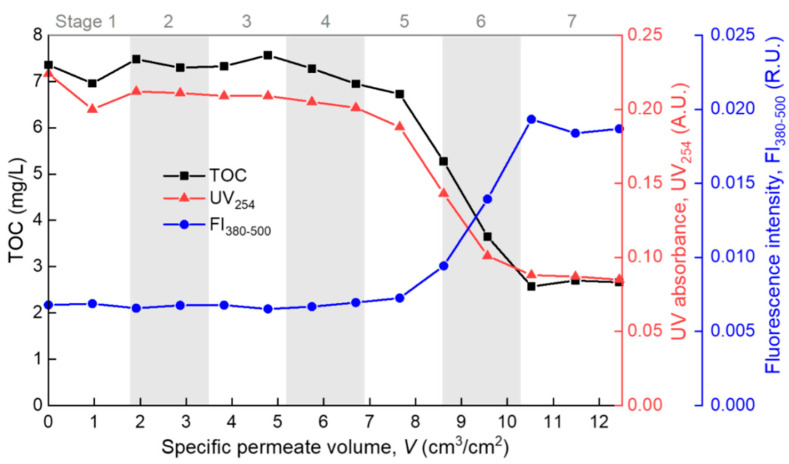
Permeate properties during the filtration process.

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
