# Peer review of "A Simple Method to Identify the Dominant Fouling Mechanisms during Membrane Filtration Based on Piecewise Multiple Linear Regression"

_membranes, 2020, doi:10.3390/membranes10080171_

Round 1
Reviewer 1 Report
This paper is quite good and I read it with a pleasure. But to be honest, it is another paper devoted to fouling mechanism and modeling it. The number of references seems to be low - I know many other paper that should be cited in such a paper.
This paper presents comprehensive approach - it is an advantage, but the rest is not new. I have a problem to indicate the novelty of this paper.
The paper is good written and well organised. The experiments were conducted in a good way. It is hard to make some critical remarks, but my overall impression is moderate (maybe a little bit higher than moderate).
Author Response
Reply:
Thanks for the positive comment.
Q1: The number of references seems to be low.
Reply:
- To make the literature review in the introduction section more complete, the following supplement is made to increase the number of references (Line 72):
- Huang, B.; Gu, H.; Xiao, K.; Qu, F.; Yu, H.; Wei, C. Fouling Mechanisms Analysis via Combined Fouling Models for Surface Water Ultrafiltration Process. Membranes (Basel) 2020, 10, doi:10.3390/membranes10070149.
- Iritani, E.; Katagiri, N. Developments of Blocking Filtration Model in Membrane Filtration. KONA Powder and Particle Journal 2016, 33, 179-202, doi:10.14356/kona.2016024.
- Giglia, S.; Straeffer, G. Combined mechanism fouling model and method for optimization of series microfiltration performance. Journal of Membrane Science 2012, 417, 144-153, doi:10.1016/j.memsci.2012.06.026.
- Kirschner, A.Y.; Cheng, Y.-H.; Paul, D.R.; Field, R.W.; Freeman, B.D. Fouling mechanisms in constant flux crossflow ultrafiltration. Journal of Membrane Science 2019, 574, 65-75, doi: 10.1016/j.memsci.2018.12.001.
- Katsoufidou, K.; Yiantsios, S.G.; Karabelas, A.J. A study of ultrafiltration membrane fouling by humic acids and flux recovery by backwashing: Experiments and modeling. Journal of Membrane Science 2005, 266, 40-50, doi: 10.1016/j.memsci.2005.05.009.
- Fernández, X.R.; Rosenthal, I.; Anlauf, H.; Nirschl, H. Experimental and analytical modeling of the filtration mechanisms of a paper stack candle filter. Chemical Engineering Research and Design 2011, 89, 2776-2784, doi: 10.1016/j.cherd.2011.04.020.
- Nakamura, K.; Orime, T.; Matsumoto, K. Response of zeta potential to cake formation and pore blocking during the microfiltration of latex particles. Journal of Membrane Science 2012, 401-402, 274-281, doi: 10.1016/j.memsci.2012.02.013.
Q2: This paper presents comprehensive approach - it is an advantage, but the rest is not new. I have a problem to indicate the novelty of this paper.
Reply:
- This paper introduces a simpler comprehensive method and the novelty is as follows:
- More complete coverage of possible mechanisms. The model incorporates five pore-blocking/cake-filtration mechanisms: 1st-kind standard blocking (fast adsorption), 2nd-kind standard blocking (slow adsorption), complete blocking, intermediate blocking and cake filtration.
- More complete span of the filtration/fouling process. The fitting clearly profiles the evolution of key mechanisms during full fouling process. However, some scholars only studied the early stages of membrane fouling.
- Statistically more rigorous. As for fitting, compared with fitting the whole curve used in previous studies, this paper adopted piecewise model fitting and the most reasonable number of segments is obtained by statistical method.
- More in line with the actual complex system. Most of the previous studies used BSA solution, while this paper we used BSA + HA + SA + inorganic components and the polysaccharides/humics/proteins proportion was similar to that of membrane bioreactor supernatants in municipal wastewater treatment.
- To make the novelty clearer to the readers, we have now added the following words to the Introduction section: “In comparison with the previous models, this method has the merits of more complete inclusion of possible mechanisms, more complete coverage of fouling stages, more realistic handling of complex fouling caused by multiple foulants, and being statistically more rigorous” (Lines 92-95 in the revised manuscript).

Reviewer 2 Report
Review of the article entitled
A simple method to identify the dominant fouling mechanisms during
membrane filtration based on piecewise multiple linear regression
Authors: Hao Xu, Kang Xiao, Jinlan Yu, Bin Huang, Xiaomao Wang, Shuai Liang, Chunhai Wei, Xianghua Wen, Xia Huang
In the presented paper the Authors proposed a simple model for identifying the main fouling mechanisms during the microfiltration process. The subject of the manuscript is within the scope of Membranes journal. The paper is consistent and well organized. Selected results are mostly supported by Authors’ explanations. I recommend to publish the manuscript after minor revision. Detailed comments are below.
It is worth considering to incorporate in the Introduction section various kinds of fouling and emphasize that the Authors considered organic fouling.
Moreover, the Authors should briefly develop the main ideas from the recalled papers to clearly show the novelty of their proposed investigations.
The Authors should put the information on the reference solution (clean water?).
What was the repeatability of the experiments?
In section 4.3. the Authors interpreted the transition of major mechanisms according to foulant composition. Can the model proposed by Authors be adapted to other foulants with smaller diameters?
Author Response
Reply:
Thanks for the positive comment.
Q1: It is worth considering to incorporate in the Introduction section various kinds of fouling and emphasize that the Authors considered organic fouling.
Reply:
- We have now added the following words in the Introduction section to emphasize the importance of organic fouling: “Membrane fouling includes organic fouling, inorganic fouling and biofouling. Organic matters do not only cause organic fouling which is prevalent in MF and UF systems, but are also seriously involved in inorganic-organic combined fouling (such as impervious gel layers due to metal-organic complexation) and bio-organic fouling (such as biofilms stubborn against cleaning). Among organic foulants, polysaccharides, proteins and humic acids are the most reported” (Lines 41-46 in the revised manuscript), with the following references added:
- Zularisam, A.W.; Ismail, A.F.; Salim, R. Behaviours of natural organic matter in membrane filtration for surface water treatment — a review. Desalination 2006, 194, 211-231, doi: 10.1016/j.desal.2005.10.030.
- Xu, H.; Xiao, K.; Wang, X.; Liang, S.; Wei, C.; Wen, X.; Huang, X. Outlining the Roles of Membrane-Foulant and Foulant-Foulant Interactions in Organic Fouling During Microfiltration and Ultrafiltration: A Mini-Review. Frontiers in Chemistry 2020, 8, doi: 10.3389/fchem.2020.00417.
Q2: Moreover, the Authors should briefly develop the main ideas from the recalled papers to clearly show the novelty of their proposed investigations.
Reply:
- In the Introduction section, we summarize the main research contents of previous papers, and states the shortcomings of their research, and then puts forward the research contents of this paper. This paper introduces a simpler comprehensive method and the novelty is as follows:
- More complete coverage of possible mechanisms. The model incorporates five pore-blocking/cake-filtration mechanisms: 1st-kind standard blocking (fast adsorption), 2nd-kind standard blocking (slow adsorption), complete blocking, intermediate blocking and cake filtration.
- More complete span of the filtration/fouling process. The fitting clearly profiles the evolution of key mechanisms during full fouling process. However, some scholars only studied the early stages of membrane fouling.
- Statistically more rigorous. As for fitting, compared with fitting the whole curve used in previous studies, this paper adopted piecewise model fitting and the most reasonable number of segments is obtained by statistical method.
- More in line with the actual complex system. Most of the previous studies used BSA solution, while this paper we used BSA + HA + SA + inorganic components and the polysaccharides/humics/proteins proportion was similar to that of membrane bioreactor supernatants in municipal wastewater treatment.
- To make the novelty clearer to the readers, we have now added the following words to the Introduction section: “In comparison with the previous models, this method has the merits of more complete inclusion of possible mechanisms, more complete coverage of fouling stages, more realistic handling of complex fouling caused by multiple foulants, and being statistically more rigorous” (Lines 92-95 in the revised manuscript).
Q3: The Authors should put the information on the reference solution (clean water?).
Reply:
- The reference solution refers to the salt background solution in this paper, which is introduced in detail in Section 3.1. To avoid ambiguity, we have unified the term into the complete name of “salt background solution” (rather than “salt background” or “background solution”), and clarified the compostion of the salt background solution as: 2 mM CaCl2, 1 mM MgCl2, 2 mM NaHCO3, 12 mM NaCl and 0.2 mM Na2SiO3 (Lines 160-179 in the revised manuscript).
Q4: What was the repeatability of the experiments?
Reply:
- According to the two repeated tests that have been done, the relative error of all the corresponding J/J0 data is within 5% (<4.4% at any V), so the error bar is omitted for convenience of presentation, and the data in this paper are all given as average values. (It has been supplemented in the caption of Figure 3 in the revised manuscript). In addition, the focus of this paper is on modeling methodology, and has little relation with data reproducibility actually.
Q5: In section 4.3, the Authors interpreted the transition of major mechanisms according to foulant composition. Can the model proposed by Authors be adapted to other foulants with smaller diameters?
Reply:
- To illustrate this problem, add a paragraph at the end of Section 4.3: “This comprehensive model, encompassing five mechanisms for pore blockage and cake filtration, should be widely applicable to various situations of dead-end filtration. For example, when the particle size is much smaller than the pore size, standard blocking may be more dominant in the model. In the case of small particle size and weak hydrophobicity, the blockage may fall into the regime of the 2nd-kind standard blocking which is slow but lasts long. If the particle size is comparable to the pore size, complete blocking may occur rapidly. Moreover, the evolution to gel layer stage may be postponed or advanced given different hardness ion concentrations for metal-organic complexing gel layer formation. These situations are all within the scope of the comprehensive model” (Lines 311-319 in the revised manuscript).

Reviewer 3 Report
This problem is relevant for journal scope. The concept and aim are clearly defined. The manuscript is well written, I could not find typing errors. The manuscript follows the formal regulations of journal.
I suggest the acceptance after major revision.
Remarks, suggestions, questions
- Please cite more papers from this journal at the last two years in the similar topic of this research.
- Please make the abbreviation list.
- What was the feed pressure in the case of experiments?
- Please add more information about model accuracy! Improve the demonstration of model verification with “precision parameters”, e. g. least squares method.
- Concentration polarization is a well-known phenomenon in the case of pressure-driven membranes. What do you think about concentration polarization in the case of your model?
- What do you think about the chance for industrial application of your work?
- Please calculate and evaluate further descriptive methods: separation factor and retention.
Author Response
Reply:
Thanks for the positive comment.
Q1: Please cite more papers from this journal at the last two years in the similar topic of this research.
Reply:
- Thanks for reminding us. The following latest relavent references have been added to the Introduction section in the revised manuscript:
- Obotey Ezugbe, E.; Rathilal, S. Membrane Technologies in Wastewater Treatment: A Review. Membranes (Basel) 2020, 10, doi:10.3390/membranes10050089.
- Yang, J.; Monnot, M.; Ercolei, L.; Moulin, P. Membrane-Based Processes Used in Municipal Wastewater Treatment for Water Reuse: State-Of-The-Art and Performance Analysis. Membranes 2020, 10, 131, doi:10.3390/membranes10060131.
- Huang, B.; Gu, H.; Xiao, K.; Qu, F.; Yu, H.; Wei, C. Fouling Mechanisms Analysis via Combined Fouling Models for Surface Water Ultrafiltration Process. Membranes (Basel) 2020, 10, doi:10.3390/membranes10070149.
Q2: Please make the abbreviation list.
Reply:
- Thanks for reminding us. The abbreviation list (in alphabetical order) has been added to the end of the manuscript (Lines 537-560):
Abbreviation list
BSA Bovine serum albumin
DOM Dissolved organic matter
DW Durbin-Watson
EEM Excitation-emission matrix
FI Fluorescence intensity
HA Humic acid
J Filtration flux
k Model coefficient (Equations 1-4)
MF Microfiltration
N Characteristic exponent
n Number of segments of the filtration process
P Trans-membrane pressure
pDW p value of DW test
R Filtration resistance
R.U. Raman unit
SA Sodium alginate
TOC Total organic carbon
t Filtration time
UF Ultrafiltration
UV Ultraviolet absorbance
V Specific permeate volume
ε Error term
μ Dynamic viscosity of the permeate
Q3: What was the feed pressure in the case of experiments?
Reply:
- Thanks for reminding us. The pressure was 5 kPa. Now this information has been added to Section 3.2 of the revised manuscript.
Q4: Please add more information about model accuracy! Improve the demonstration of model verification with “precision parameters”, e. g. least squares method.
Reply:
- Thanks for reminding us. Yes, we used least-squares regression in the “stepwise” mode for the fitting, which has now been clarified in Line 128 and Lines 150-151 in the revised manuscript. Then R-squared was employed as a dimensionless goodness-of-fit measure of the regression models. The R-squared values have now been attached to the captions of Figures 3 and 4, such as “R-squared values for the fitting in stages 1-7 are 0.983, 0.990, 0.985, 0.983, 0.996, 0.980 and 0.987, respectively”.
Q5: Concentration polarization is a well-known phenomenon in the case of pressure-driven membranes. What do you think about concentration polarization in the case of your model?
Reply:
- Concentration polarization (CP) stems from mechanical rejection of the foulant by the membrane. It creates a concentration gradient that repels the foulant moving toward the membrane, i.e. hinders mass transfer due to chemical potential gradient. Unlike that in reverse osmosis processes, CP in microfiltration (MF) and ultrafiltration (UF) processes only involves colloids and macromolecules (rather than salt ions), and hence has little direct contribution to increase in osmotic pressure, transmembrane pressure and filtration resistance. This suggests that, in MF and UF systems, CP influence the filtration resistance implicitly (via mass transfer effect) rather than intervene in the pore blocking or cake filtration modes explicitly. Therefore, in the present study, the influence of CP is not considered in our model; if there is any, it is asumed to be included in the error term of the model. For clarity, we have now added the following words to Section 2 Model: “…The error term, ε, encompasses errors due to random error of V and J, deviation of N (as a result of e.g. non-uniformity of the pore structure or polydispersity of the foulant particles), neglect of concentration polarization (a feedback effect on foulant mass transfer) and other fouling mechanisms, and possible interactive influences between different mechanisms (e.g. at adjacent areas of the membrane) that cause nonlinearity of Equation 4…” (Lines 118-123 in the revised manuscript).
Q6: What do you think about the chance for industrial application of your work?
Reply:
- Thanks for reminding us. Regarding this, we have added a short comment to the end of the manuscript: “…(this method) also provides a tool to assess membrane fouling potential, characterize foulant properties and understand membrane-foulant interactions, which will profoundly support optimal selection of membrane and targeted pretreatment of foulant solution for efficient fouling control in industrial applications” (Lines 333-336 in the revised manuscript).
- Additionaly, considering the complexity of foulants in real water/wastewater, we have also added the following comments to Section 4.3: “This comprehensive model, encompassing five mechanisms for pore blockage and cake filtration, should be widely applicable to various situations of dead-end filtration. For example, when the particle size is much smaller than the pore size, standard blocking may be more dominant in the model. In the case of small particle size and weak hydrophobicity, the blockage may fall into the regime of the 2nd-kind standard blocking which is slow but lasts long. If the particle size is comparable to the pore size, complete blocking may occur rapidly. Moreover, the evolution to gel layer stage may be postponed or advanced given different hardness ion concentrations for metal-organic complexing gel layer formation. These situations are all within the scope of the comprehensive model” (Lines 311-319 in the revised manuscript).
Q7: Please calculate and evaluate further descriptive methods: separation factor and retention.
Reply:
- Following the suggestion, we have calculated the retention rate of the membrane during the filtration process, and added the following words to Section 4.3: “…Correspondingly, the TOC rejection rate was below 10% at pre-cake stages (stages 1-4) but grew to over 60% at the cake/gel layer stage (stage 7)...” (Lines 294-295 in the revised manuscript).

Round 2
Reviewer 3 Report
I suggest the acceptance in this present form for publication.